# Surface Optimization of Components Obtained by Fused Deposition Modeling for Air-Plasma-Sprayed Ceramic Coatings

Antonio Formisano *, Luca Boccarusso and Antonio Langella

Department of Chemical, Materials and Production Engineering, University of Naples Federico II, P.le V. Tecchio 80, 80125 Napoli, Italy; luca.boccarusso@unina.it (L.B.); antgella@unina.it (A.L.)
* Correspondence: aformisa@unina.it; Tel.: +39-081-768-5208

**Abstract:** Additive manufacturing is an emerging disruptive 3D printing technology that is stimulating innovations in design and engineering, materials, and manufacturing thanks to the prospects of reducing cost and waste and increasing efficiency; in doing so, it presents the potential to have marked industrial, economic, and societal impacts. Thermoplastic polymers show some ideal characteristics for the most common additive manufacturing methods, and this aids in the improvement of the design accuracy and reliability and makes inroads for the customized manufacturing of high-design-flexibility polymer parts. Despite this, this material family is strongly sensitive to temperature, and one of the viable ways of limiting this weak point is surface coating with thermal barriers. The focus of this work was the optimization of an additive manufacturing process for producing thermoplastic components and to improve the adhesion of a thermal barrier coating on their surface. In detail, flat plate specimens of ULTEM 1010 were obtained by the fused deposition modeling technique by varying two significant surface parameters, the enhanced visible rasters and the visible raster air gap; then, their surfaces were covered by a thin ceramic coating by an air plasma spray system. A micro-geometric analysis that was conducted using a confocal microscope and the coating thickness measurements highlighted that a global larger roughness value, the presence of more flat peaks with a large area, and the complexity of the texture can be considered as supporting factors for improving the mechanical gripping and allowing a uniform adhesion of the coating powders on the thermoplastic substrate.

**Keywords:** fused deposition modeling; thermoplastic polymers; air plasma spray; thermal barrier; surface optimization

## 1. Introduction

Additive manufacturing (AM) is a relatively recent fabrication technique that shows, as strong points, the ability to manufacture custom-oriented and complex components, minimizing waste and consumption; thanks to its numerous advantages, AM has become a very popular trend in manufacturing processes. Some of the businesses that use AM involve prototyping, building, and biomechanical engineering, as well as the aerospace and automotive sectors, due to the unrivalled freedom of manufacturing design that this 3D printing technology allows [1]. The process, developed by Hull in 1986 as stereolithography [2], involves the building of parts, whose 3D solid models are developed and converted into an AM file format, layer by layer (material deposed on top of each other) by an AM machine. As a consequence of the expiry of the earlier patents, this process has become more and more popular; an increasing number of AM materials and methods are being developed to meet the demand of printing complex components with fine resolutions, reducing printing defects, and guaranteeing increased mechanical properties [3], and their drawbacks and benefits have been discussed [4]. A very common method of 3D printing is fused deposition modeling (FDM), which mainly uses polymer filaments. The other main AM methods include, but are not limited to, selective laser sintering (SLS), selective

laser melting (SLM), inkjet printing, direct energy deposition (DED), and laminated object manufacturing (LOM).

As anticipated above, FDM is probably the most widely used among the various methods of 3D printing [5]. It is a material-extrusion-based AM technique with which products are fabricated by melting filaments and depositing molten materials on a platform [6]. In more detail, this technique involves the use of compact equipment with low maintenance costs; a movable head extrudes and deposes the material in ultra-thin layers onto a substrate. The material, as soon as deposited, solidifies and cold-welds to the previous layers [7]. FDM largely uses thermoplastics, but the use of wax, metals and ceramics is being considered [8].

ULTEM 1010 is an amorphous polyetherimide thermoplastic polymer, showing flame-retardant properties and high thermal stability, which make it an ideal resin for the biomedical field [9], as well as for use in out-of-cabin aerospace and under-the-hood automotive applications [10]; it is largely used for FDM since it represents a material that can opportunely be modified with functional additives and processed into filaments [11]. The behavior of ULTEM 1010 samples obtained by FDM is strongly influenced by the process parameters, so much so that it is advisable to investigate their weight through a rigorous and extensive testing campaign. By way of example, full-factorial design of experiments (DOE) was considered to investigate the influence of the build orientation and raster angle on the flexural response [12] and fracture toughness [13] of FDM solid-build ULTEM 1010 specimens.

Polymer-based components are extremely sensitive to temperature, both when machined and [14,15] and under working conditions [16]; they can be protected from the effects of heat, hot gases, and fire by using opportune thermal barrier coatings (TBCs) [17]. TBC deposition represents a crucial thermal insulation technology because it enables the underlying substrate to operate near or above its melting temperature. TBCs can be fabricated by various processing methods; all of them pursue the goal of the development of microstructures with desired control over thermal conductivity and mechanical properties, as well as process adaptability and affordability [18]. Despite the innovative techniques that are used for specific properties and applications, two traditional methods still represent the most widely used methods for the deposition of TBCs, i.e., air plasma spray (APS) and electron beam–physical vapor deposition (EB-PVD). Regarding the first one, APS has strong points such as having low costs, a rapid deposition rate, a high efficiency, it is easily manageable, and it can be applied to all suitable base materials with the widest variety of powders [19]. Composite ceramic TBCs are largely used in aerospace due to their excellent thermal insulation; thanks to them, it is possible to improve significantly the service life and durability of the coated components [20]. One of the most successful TBC materials is the yttria-stabilized zirconia/MCrAlY (M = Ni, Co) two-layer structure; it presents a bond coat, providing oxidation and corrosion resistance, as well as the matching of thermal properties and stress between the substrate and the ceramic coating, and a ceramic top coat with a low thermal conductivity, providing thermal insulation and thermal shock resistance [21].

The preparation of the substrate of the components to be coated is crucial for the optimal adhesion of the TBC material particles [22]; in fact, the surface properties can significantly influence the formation of mechanical bonds between the impact particles and the external surfaces of the components acting as substrate [23]. Concerning this, an adequate micro-geometric characterization can be very useful for revealing deeper insights into 3D printing about, for example, the morphology of the surfaces and the presence of defects; 3D surface measurements and characterization give a better understanding of the surfaces in their functional state, thus overcoming the limitations of 2D techniques. To do this, different methods can be considered. Among them, the X-ray computed tomography (XCT) scanning methods [24,25] furnish image-based models that, as real reflections of the material's micro/mesostructures, offer tremendous potential for investigating damage, fractures, and failure mechanisms with a higher accuracy [26]. At the same time, several spectroscopic techniques, including confocal laser scanning microscopy (CLSM), Fourier

transform infrared (FTIR) spectroscopy and photoacoustic FTIR spectroscopy, Raman spectroscopy, and X-ray photoelectron spectroscopy (XPS) are widely used for investigating 3D-printed parts obtained by FDM. Among them, the CLSM method was used, for example, for the observation of polylactic acid (PLA) specimens after tensile tests in [27] to investigate the fracture planes or for the evaluation of the impact of the FDM process on PLA chemistry and structure [28].

Considering the above, the present work aimed to investigate the influence of two significant surface parameters of an FDM process on ULTEM 1010 samples and to promote the adhesion of the impacting particles of a ceramic composite material. Flat square thermoplastic plates were manufactured by varying the two investigated process parameters and, subsequently, were covered by a ceramic TBC by an APS system. The micro-geometric analysis of the samples' surfaces (before and after APS deposition), employing a CLSM device [29] and through a methodical interpretation of the different morphological surface parameters, and the evaluation of the TBC thicknesses allowed a correlation of the goodness of depositing with the surface parameters of the AM technique to be obtained.

## 2. Materials and Methods

In this work, the FDM process of solid square plates, $50 \times 50 \times 5$ mm$^3$ in size, was conducted using a Stratasys Fortus 450 mc, which is a 3D printer that can print high-performance materials with an accuracy of $\pm 0.127$ mm. It presents the extrusion head with two nozzles, specific for each pair of materials (model and support). The process was controlled by a PLC and the Stratasys Insight 16.11 application software for the preparation of the CAD program's STL output for 3D printing. The main properties of ULTEM 1010, the thermoplastic material processed by FDM, were taken from the Stratasys data sheets and are reported in Table 1 [10]. As expected, the mechanical properties depended on the setting of the FDM process parameters, and, in addition, the specimens created using FDM showed anisotropy because of the printing process; the influence of the process parameters on different mechanical properties has been investigated in many research works [12,30,31]. Then, the values reported in the table must be considered reference values.

**Table 1.** Main properties of ULTEM 1010 [10].

| Mechanical properties | |
|---|---|
| Tensile strength, Yield (ASTM D638) [MPa] | 64 (XZ) 41 (ZX) |
| Tensile strength, Ultimate (ASTM D638) [MPa] | 81 (XZ) 48 (ZX) |
| Tensile modulus (ASTM D638) [GPa] | 2.77 (XZ) 2.20 (ZX) |
| Tensile elongation at break (ASTM D638) [%] | 3.3 (XZ) 2.0 (ZX) |
| IZOD impact, Notched (ASTM D256) [J/m] | 41 (XZ) 24 (ZX) |
| Compressive strength, Yield (ASTM D695) [MPa] | 134 (XZ) 107 (ZX) |
| Compressive strength, Ultimate (ASTM D695) [MPa] | No break (XZ) 1.125 (ZX) |
| Compressive modulus (ASTM D695) [GPa] | 10.00 (XZ) 1.12 (ZX) |

**Table 1.** *Cont.*

| Thermal properties | |
| --- | --- |
| Heat deflection at 264 psi (ASTM D648) [°C] | 213 |
| Glass transition temperature (DSC) [°C] | 215 |
| Coefficient of thermal expansion (ASTM E831) [μm/(m·°C)] | 47 |
| **Other** | |
| Specific gravity (ASTM D792) [g/cm$^3$] | 1.27 |

There are several FDM process parameters related to surface quality, contour, and material consumption. However, since the surface quality results were very important in this study, the influence of only two surface parameters as control factors was considered because they directly affected it, namely enhanced visible rasters (EVR) and visible raster air gap (VRAG). They represented the infill line width and the raster-to-raster gap for the visible up-facing surfaces, respectively (see Figure 1). For both, three levels were considered. To determine the choice ranges of these values, a series of preliminary tests were carried out, starting from the extreme values that could be entered into the software and conducting a visual inspection of the 3D printing surface quality; Figure 2 shows examples of bad (a) and good (b) qualities of the printed samples. Table 2 summarizes the main process parameters and the related values; for the two control factors, the three levels were labelled as −1, 0, and 1, whereas the other parameters were held constant.

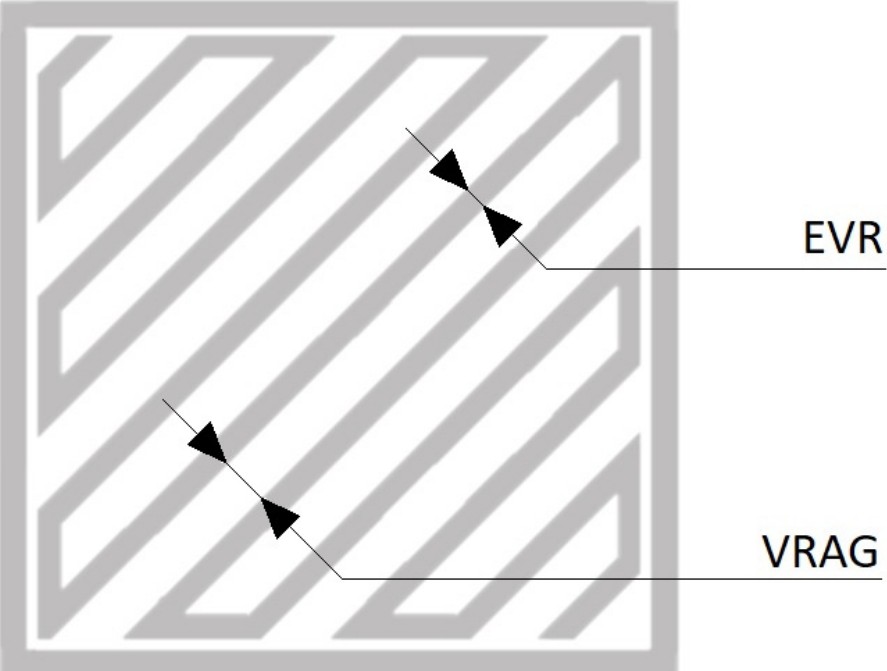

**Figure 1.** Schematization of enhanced visible rasters (EVR) and visible raster air gap (VRAG).

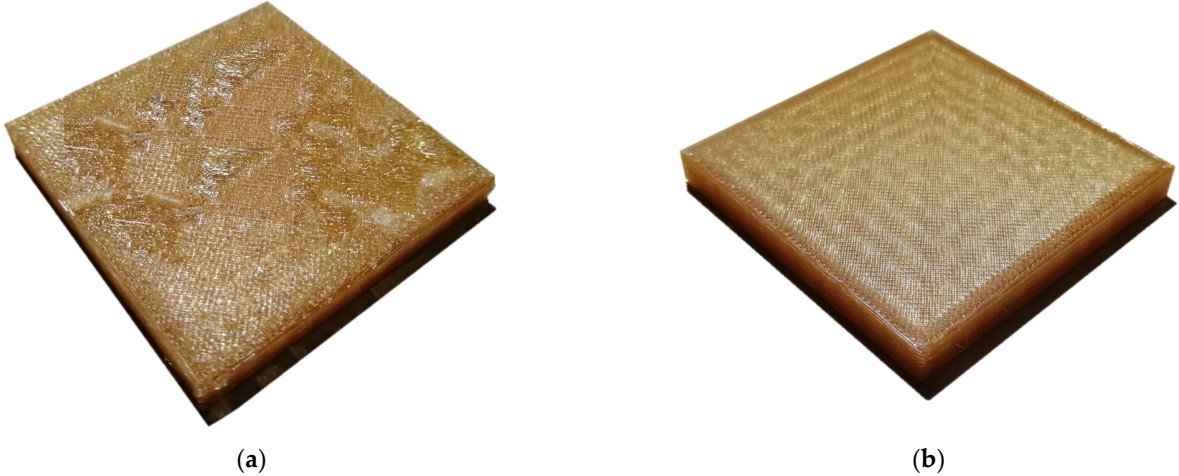

(**a**)                                                    (**b**)

**Figure 2.** Preliminary 3D printing tests: (**a**) bad and (**b**) good surface quality results.

**Table 2.** FDM process parameters.

| Parameter | Value | | |
|---|---|---|---|
| Enhanced visible rasters [mm] | 0.3556 | 0.5431 | 0.7306 |
| Visible raster air gap [mm] | −0.0254 | 0.0381 | 0.1016 |
| Enhanced internal rasters [mm] | | 0.3556 | |
| Internal raster air gap [mm] | | 0 | |
| Raster angle [°] | | 45 | |
| Contour width [mm] | | 0.3556 | |
| Number of contours | | 2 (On the visible surface, the default single contour was used) | |

A face-centered central composite design was chosen for the execution of the FDM process; a total of twenty-seven experimental runs were carried out with three replicates for each of the nine experimental setups. Table 3 summarizes the experimental design; the different setups were labelled from I to IX. The samples were printed with a randomized layout to avoid correlations between the printing results and positioning on the printing table.

**Table 3.** Experimental design of the 3D-printing process.

| Setup | EVR | VRAG |
|---|---|---|
| I | −1 | −1 |
| II | 0 | −1 |
| III | 1 | −1 |
| IV | −1 | 0 |
| V | 0 | 0 |
| VI | 1 | 0 |
| VII | −1 | 1 |
| VIII | 0 | 1 |
| IX | 1 | 1 |

The second step was the deposition of the ceramic coatings on the flat surface of the FDM components. An APS system was used, setting standard parameters, for the deposition of a coating with a nominal thickness of 0.6 mm. The TBC was composed of a topcoat (nominal thickness of 0.4 mm), i.e., ceramic layers of yttria-stabilized zirconia ($ZrO_2$-$Y_2O_3$ 93-7) deposited on a bond coat of NiCrAlY (nominal thickness of 0.2 mm), which was aimed to improve the adhesion of the ceramic layer as well as to provide an antioxidant/protective shield for the substrate. The topcoat powder (supplied by Amperit) had a spherical, porous or hollow, partly open morphology (HOSP) with a nominal size

distribution of 10–125 μm, whereas the bond powder (supplied by Oerlikon Metco) had a spheroidal morphology with a nominal size distribution of 53–106 μm. The spray gun of the APS system deposited the coat moving up and down along an axis parallel to that of a rotating support. It presented eight housing slots on which the metal plates were adapted; up to four samples could be fixed on each plate according to a randomized arrangement to house and process all the samples together. Figure 3 reports a schematization of the APS deposition.

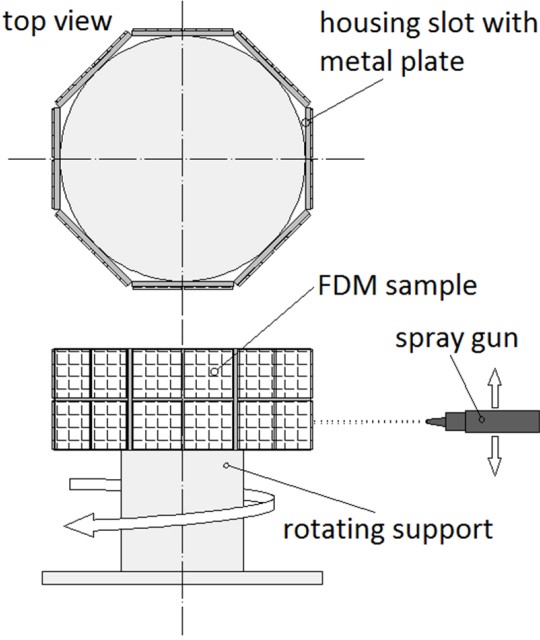

**Figure 3.** Schematization of the APS deposition.

The micro-geometric characterization of the up-facing surfaces of the FDM samples by the measuring of height, feature, hybrid, and areal parameters was carried out through a Leica DCM 3D confocal microscope equipped with the LeicaScan and LeicaMap softwares (standard versions) for managing the various modes and configuration options for the confocal images and for their post-processing analysis, respectively; the nomenclature of the morphological surface parameters investigated is reported in Table 4. A surface of $32 \times 3.96$ mm$^2$ was scanned in the central area of each sample with a $10\times$ objective magnification, a $3\times$ scanning speed, and an 800 μm asymmetric Z scanning; this represented a good compromise between the scanning time and the acquired surface size. The square-shaped samples were manually positioned on the microscope support, and a reference adhesive tape was fixed on the support to allow greater precision and alignment. All the morphological surface parameters indicated by the ISO 25,178 standard as the best representatives of the phenomenon of the adhesion of powders by plasma spray technology were selected. They were:

**Table 4.** Nomenclature of the morphological surface parameters.

| Nomenclature | Morphological Surface Parameter |
|:---:|:---|
| $S_a$ | Arithmetic mean height (height parameter) |
| $S_q$ | Root mean square height (height parameter) |
| $S_{ku}$ | Kurtosis (height parameter) |
| $S_{pd}$ | Number of peaks per unit area (feature parameter) |
| $S_{dr}$ | Developed interfacial area ratio (hybrid parameter) |
| $P_A$ | Peak area (areal parameter) |
| $D_A$ | Dale area (areal parameter) |

- Arithmetic mean height, $S_a$—this is defined as the arithmetic mean of the absolute value of the height, $Z(x,y)$, within a sampling area ($A$):

$$S_a = \frac{1}{A} \int \int_A |Z(x,y)| dx dy \tag{1}$$

- Root mean square height, $S_q$—this is the root mean square value of the surface departures within the sampling area:

$$S_q = \sqrt{\frac{1}{A} \int \int_A Z^2(x,y) dx dy} \tag{2}$$

- Kurtosis, $S_{ku}$—this is a measure of the sharpness of the surface height distribution and is the ratio of the mean of the fourth power of the height values and the fourth power of $S_q$ within the sampling area:

$$S_{ku} = \frac{1}{S_q^4} \left[ \frac{1}{A} \int \int_A Z^4(x,y) dx dy \right] \tag{3}$$

Kurtosis is strictly positive and unitless and characterizes the spread of a height distribution. A surface with a Gaussian height distribution has a kurtosis value of three. The use of this parameter not only detects whether profile spikes are evenly distributed but also provides a measure of the spikiness of an area. A spiky surface has a high kurtosis value, and a bumpy surface has a low kurtosis value.

- Number of peaks per unit area, $S_{pd}$;
- Developed interfacial area ratio, $S_{dr}$—this is expressed as the percentage of the additional surface area contributed by the texture as compared to an ideal plane with the size of the measurement region:

$$S_{dr} = \frac{(Texture\ Surface\ Area) - (Cross\ Sectional\ Area)}{Cross\ Sectional\ Area} \cdot 100 \tag{4}$$

The $S_{dr}$ parameter is used as a measure of surface complexity and can provide useful information in applications involving surface coatings and adhesion. It is affected both by texture amplitude and spacing and increases with the spatial intricacy of the texture, whether or not $S_a$ changes.

In addition, two other response variables were also considered: the peak area ($P_A$) and the dale area ($D_A$). They represent the area of peaks and the dales of surfaces. Peaks represent material that is above the middle line of the height distribution, while valleys are material below this line. Concerning the coating thickness, it was calculated as the difference between the height of the samples before and after the TBC deposition. To have a reliable value, nine measurements were made on nine equally spaced points of each sample; a measuring mask was designed and 3D-printed to ensure that they were always acquired in the same points both before and after plasma spraying. Measurements were carried out using a digital height gauge with a 2 mm diameter spherical probe.

The complete research framework proposed in this work is summarized in the flowchart of Figure 4.

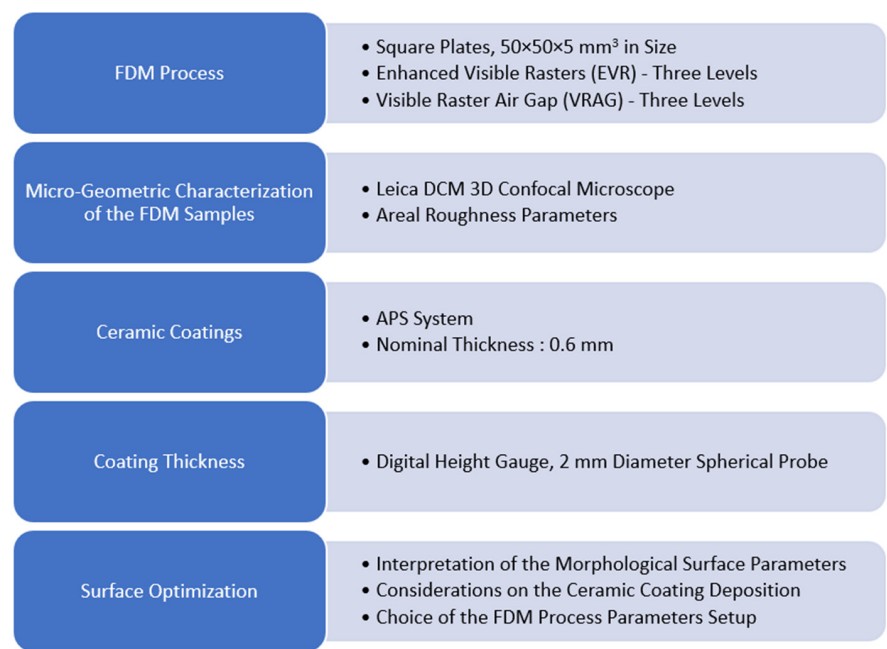

**Figure 4.** Flowchart of the research framework.

## 3. Results

Examples of the scanned FDM surfaces for some representative experimental setups (the extreme cases, i.e., the setups I and IX, and the intermediate case, i.e., the setup V) are shown in Figure 5; the figure highlights how each surface, and consequently the corresponding surface morphology, results were different due to the influence of the FDM process parameters. This aspect reflects on the different nature of APS deposition; concerning this, Figure 6 reports the three repetitions of the APS samples for each experimental setup. Finally, Table 5 summarizes the corresponding coating thicknesses (both in terms of the average and standard deviation).

**Table 5.** Coating thicknesses (average and standard deviation).

| Setup | Average [mm] | Standard Deviation [mm] |
|-------|--------------|-------------------------|
| I | 0.024 | 0.011 |
| II | 0.425 | 0.055 |
| III | 0.543 | 0.011 |
| IV | 0.501 | 0.035 |
| V | 0.514 | 0.033 |
| VI | 0.509 | 0.011 |
| VII | 0.483 | 0.012 |
| VIII | 0.488 | 0.021 |
| IX | 0.498 | 0.011 |

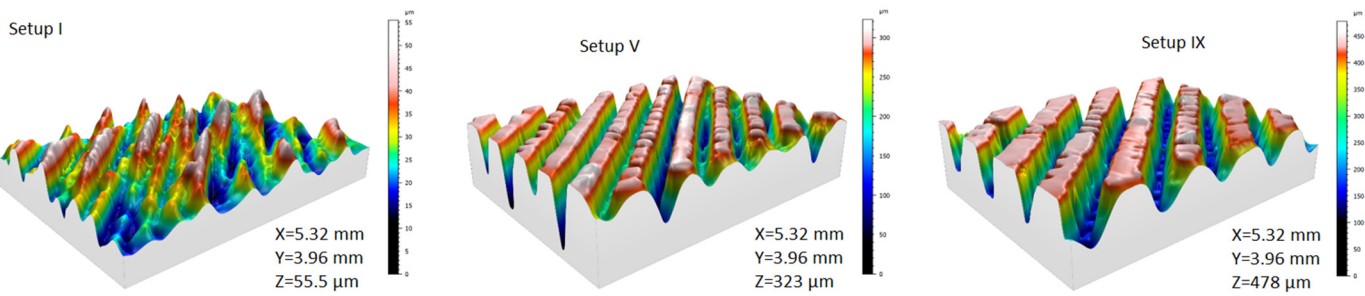

**Figure 5.** Scanned FDM surfaces for setup I (**left**), setup V (**center**), and setup IX (**right**).

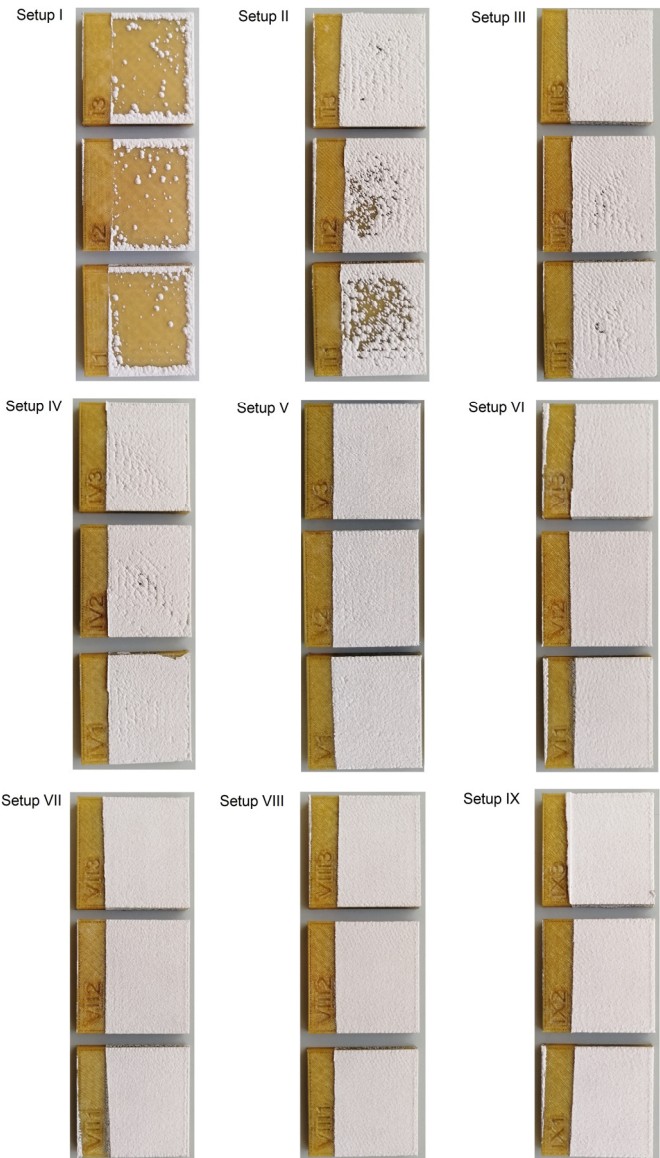

**Figure 6.** Plasma-sprayed samples for each experimental setup.

## 4. Discussion

The adhesion of metal particles to a polymer substrate was difficult due to the different natures of the chemical bonds of the two elements: metal bonds for the powders and covalent bonds for the ULTEM 1010 substrate. Consequently, the mechanical interaction between the sprayed particles and the substrate tended to prevail over other phenomena, and differences in the coating adhesion were ascribed to the substrate morphology. As a result, the morphological characteristics of the FDM polymer surface strongly influenced the coating adhesion. It was almost nil for the setup I, and the results were defective for the setup II. For the remaining ones, the coating adhered uniformly on the FDM surfaces. These aspects can be observed in Figure 6 and Table 5. To explain these results from a technological point of view, a comparison was made between the two extreme cases of the experimental campaign, i.e., setup I and setup IX, which presented no coating and a uniform coating adhesion, respectively. Figure 7 shows the scanned as-built surfaces and the APS surfaces for these two cases, while Table 6 summarizes the information on the main parameters from the microscopy.

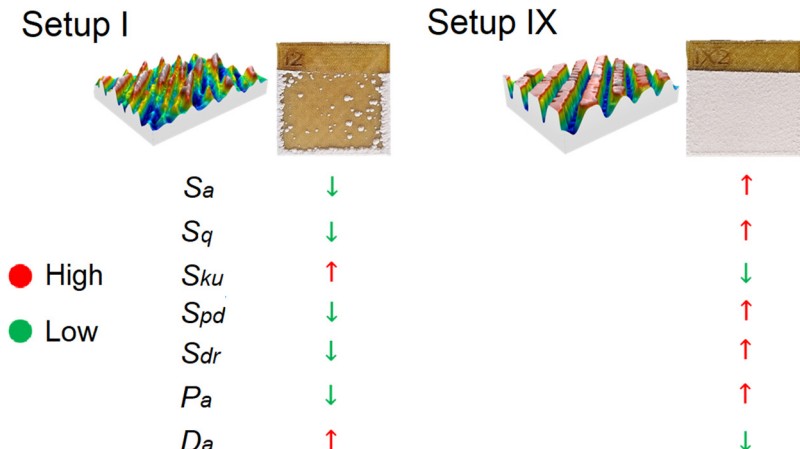

**Figure 7.** As-built surfaces and APS surfaces for setup I (**left**) and setup IX (**right**).

**Table 6.** Morphological surface parameters for setup I and setup IX.

| Morphological Surface Parameter | Setup I | Information | Setup IX | Information |
|---|---|---|---|---|
| $S_a$ [μm] | 10.39 | Low average roughness | 110.81 | High average roughness |
| $S_q$ [μm] | 2.51 | Low roughness variability | 28.33 | High roughness variability |
| $S_{ku}$ [–] | 61.28 | Spiked peaks and dales | 2.58 | Flat peaks and dales |
| $S_{pd}$ [1/mm$^2$] | 0.57 | Few peaks | 6.31 | Many peaks |
| $S_{dr}$ [%] | 18.64 | Uniform texture | 132.12 | Complex texture |
| $P_A$ [mm$^2$] | 10.65 | Smaller peaks area | 12.45 | Larger peaks area |
| $D_A$ [mm$^2$] | 10.45 | Larger dales area | 8.78 | Smaller dales area |

The absence of adhesion for setup I was ascribable to the very good surface finishing (the particles did not adhere but bounced on the substrate); the average and the variability of the roughness of setup I were far lower than those in setup IX, as pointed out by the corresponding $S_a$ and $S_q$ values. Peaks and dales were more spiked in setup I ($S_{ku} >> 3$) than in setup IX, which was characterized by peaks and dales with a flat surface ($S_{ku} < 3$). In addition, the number of peaks per unit area was also very different; few peaks were observed for setup I (low value of $S_{pd}$) and many peaks were observed for setup IX (high value of $S_{pd}$). Furthermore, setup I had a quite uniform texture (low value of $S_{dr}$) while setup IX had a complex texture (high value of $S_{dr}$). Finally, setup IX was characterized by a slightly higher $P_A$ value (and a corresponding slightly lower $D_A$ value) that permitted the particles to find a higher surface on which to adhere. Figure 8 reports the scanning of the surfaces of setup I and setup IX after the APS coating deposition (corresponding to the yellow frames in the figure), providing further confirmation of the different adhesion capacities for the two setups in a qualitative way.

For a further quantitative analysis, two profiles were extracted from the same surfaces before the deposition; Figure 9 shows them for setups I (up) and IX (down). Setup I was characterized by an average distance between the peaks and valleys of 40 μm, while for setup IX, this distance was 300 μm. For setup I, it was difficult to identify the primary peaks and to establish a horizontal distance between two of them, but either way it was much smaller than that of setup IX, which was 500 μm.

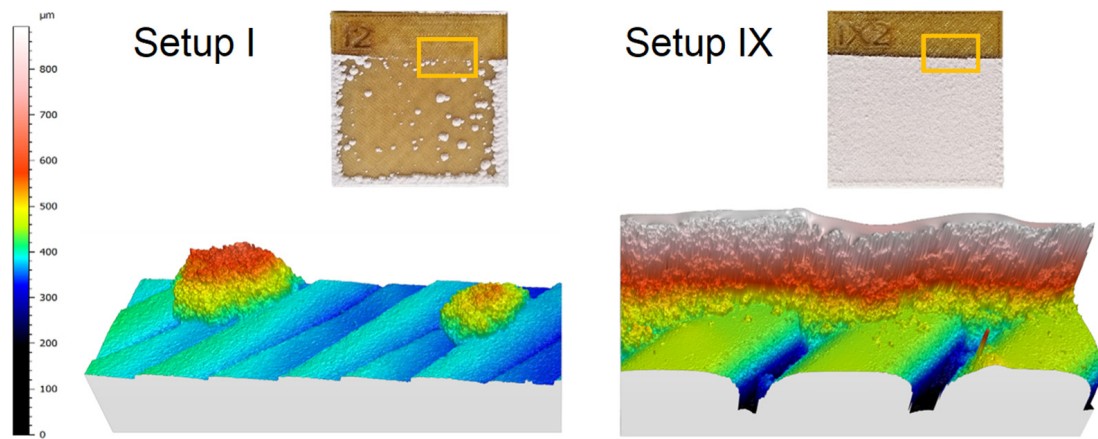

**Figure 8.** Surfaces after APS deposition for setup I (**left**) and setup IX (**right**).

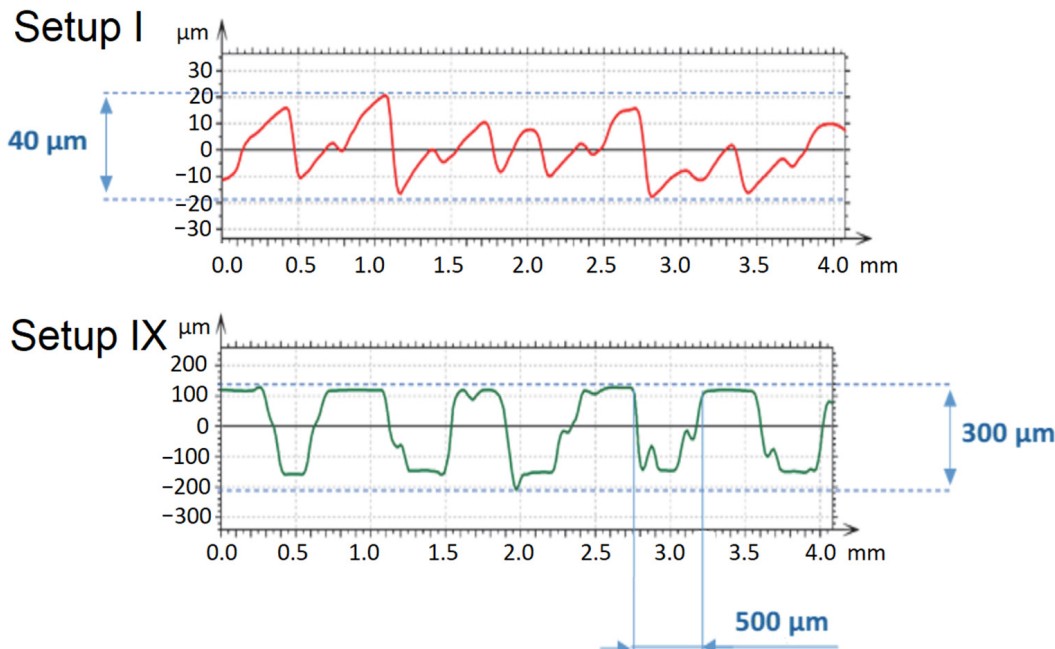

**Figure 9.** Extracted profiles from FDM surfaces for setup I (**up**) and setup IX (**down**).

By considering that, for the bond layer, the size of the sprayed metal particles varied from 53 to 106 μm, for setup IX, the powder could enter the valleys because its size was smaller than the average horizontal distance between two peaks. Similar considerations on the profiles can justify the highest coating thickness for setup III, considering a more favorable spacing between the peaks and valleys of the FDM substrate.

## 5. Conclusions

An experimental campaign was carried out to investigate the influence of two FDM surface parameters on the surface quality of flat thermoplastic samples and to promote a ceramic coating deposition by APS. The results from the experimental campaign with nine different setups highlighted that the two surface parameters, namely the enhanced visible rasters and the visible raster air gap, significantly influenced the adhesion capacity of the FDM samples. The different nature of the chemical bonds, i.e., metal bonds for the powders and covalent bonds for the ULTEM 1010 substrate, made the adhesion of the metal particles to the polymer substrate difficult and, consequently, it was fundamental to focus on their mechanical interactions based on the substrate morphology. The main results were the following:

- The adhesion of the coating to the substrate increased with its roughness;
- No coating or defective adhesion were found for setups I and II, respectively. This was a consequence of the texture produced by a very good surface finishing, which promoted the bouncing of the ceramic particles on the substrate;
- Starting from setup III, the coating was uniform on the FDM surfaces. They presented a complex texture and a surface with higher peaks that promoted the deposition;
- The coating thickness for the best setup was about 0.54 mm, and the powder could enter the valleys of the texture since they presented a smaller size compared to the horizontal distance between two consecutive peaks.

In conclusion, this work highlights that, as well as composite materials, it is fundamental to correctly design a 3D-printing process through the opportune choice of the process parameters combined with the efficient use of statistical tools while minimizing the shortcomings in the engineering practices. Further developments should be focused on, but not limited to, performing adhesion tests on sprayed samples to verify which experimental setup involves a stronger or weaker adhesion to the substrate, thus extending the investigation to other FDM parameters that may affect the achievable surface roughness of parts produced by FDM and deepening the non-linear relationship between some of the roughness parameters considered in this work and the deposition of the coatings.

**Author Contributions:** Conceptualization, A.F., L.B. and A.L.; methodology, L.B. and A.L.; investigation, L.B.; data curation, A.F.; writing—original draft preparation, A.F.; writing—review and editing, A.F. and L.B.; supervision, A.L. All authors have read and agreed to the published version of the manuscript.

**Funding:** This research received no external funding.

**Data Availability Statement:** Not applicable.

**Conflicts of Interest:** The authors declare no conflict of interest.

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
