# Peer review of "Surface Optimization of Components Obtained by Fused Deposition Modeling for Air-Plasma-Sprayed Ceramic Coatings"

_jcs, doi:10.3390/jcs7040158_

Round 1

Reviewer 1 Report

This paper presents an optimization of an additive manufacturing process of thermoplastic components, to improve the adhesion of a thermal barrier coating on their surface. This work is interesting and has been well presented. The following questions should be revised before future publication.

1. Abstract: The significance or starting point of the study should be briefly introduced at the beginning. The direct introduction of the research content of this article is a little unfriendly for readers who are not in this research field.

3. Introduction: the writing of the current research review is fine, especially for Fused Deposition Modeling and ceramic TBCs. However, the research purpose is not refined from the current literature review. In this paper, it is needed to further elaborate on the bottleneck problem in the processing of Ceramic Coatings for thermoplastic components. Polymer components are extremely sensitive to temperature. The following research (https://doi.org/10.1016/j.jmrt.2022.12.054. https://doi.org/10.1016/j.cja.2022.12.009) also mentioned this topic, which can be added to enrich your literature review.

4. The measurement the thickness of deposited coating is very difficult. Figure 6 show the average coating thickness with a test accuracy of micrometer. The size error of the surface groove of the composite before spraying will reach hundreds of microns (Fig. 4). The measuring principle of the thickness of deposited coating needs to be explained in detail. Besides, the standard deviation between round brackets can be changed to error bars in Fig. 6.

5. Improve the conclusion. The innovative results can be listed separately.

Reviewer 2 Report

This work aims to improve the adhesion of a thermal barrier coating on the surface of thermoplastic components, during the additive manufacturing process. The micro-geometric characterization of surfaces is conducted for the 3D printed samples to reveal the effects of surface parameters on the adhesion capacity. Overall, the manuscript is easy to follow and has some merits within the scope of this journal. However, it requires better introduction of the methods and deeper explanation of the results.

(1) In the introduction, the novelty needs to be more explicitly presented, i.e. what is new in this work while the other literatures have not conducted. Please present a more concise and correlated literature review. A scientific paper should stress the innovative points and not describe all information like a report. The Abstract and Conclusions lack in-depth demonstration of the quantitative results. The reviewer suggests rewriting the Conclusions in a point-by-point manner using new/important findings.

(2) The scanned FDM surfaces are interesting and attractive to potential readers. What is the scanning device? Please mention it in the context. The authors are encouraged to compare the scanning results with X-ray Computed Tomography scanning tests. In the introduction part, the manuscript should at least mention the capability of XCT-based methods, e.g., 10.1016/j.ijsolstr.2015.05.002, 10.1016/j.cemconcomp.2021.104347, and discuss how they can be utilized to reveal deeper insights into 3D printing, with regard to 3D surfaces and defects, etc.

(3) The authors are encouraged to draw a 3D printing flowchart for their materials. This can be more informative for readers and helpful to promote their methods and framework in the research community.

(4) The roughness parameters are very important and the authors should introduce their physical meanings (why use them) and how to compute them, e.g. skewness, kurtosis, developed interfacial area ratio, area of the dales/peaks, etc.?

(5) Why only Setup I and Setup IX are discussed? The reviewer thinks it is also useful to discuss Setup III with the largest mean thickness and Setup IV with the largest standard deviation. In addition, the analyses of CoV are also necessary. The reviewer is also confused as to which roughness parameter is the most important/dominant in the surface assessment.

(6) The authors should discuss the effects of surface parameters on the mechanical properties, e.g. strength and elasticity along different directions, because 3D printing can result in strong anisotropy.

(7) In the concluding part, some discussions on the limitations and future work would be helpful, for example, how to make full advantages of 3D printing while minimizing their shortcomings in engineering practices?

Round 2

Reviewer 1 Report

The review comments are replied correctly. So, the current status of the manuscript can be accepted for publication. 

Reviewer 2 Report

The manuscript has been much improved and thus deserves publication in the current form.